

# Characterization and toxicity of citral incorporated with nanostructured lipid carrier

Noraini Nordin[1], Swee Keong Yeap[2,3], Nur Rizi Zamberi[1], Nadiah Abu[1,4], Nurul Elyani Mohamad[1], Heshu Sulaiman Rahman[5], Chee Wun How[3,6], Mas Jaffri Masarudin[1,3], Rasedee Abdullah[3] and Noorjahan Banu Alitheen[1,3]

[1] Department of Cell and Molecular Biology, Faculty of Biotechnology and Biomolecular Sciences, Universiti Putra Malaysia, Serdang, Selangor, Malaysia

[2] China-ASEAN College of Marine Sciences, Xiamen University Malaysia, Sepang, Malaysia

[3] Institute of Bioscience, Universiti Putra Malaysia, Serdang, Selangor, Malaysia

[4] UKM Medical Centre, UKM Medical Molecular Biology Institute (UMBI), Cheras, Wilayah Persekutuan, Malaysia

[5] Department of Microbiology, Faculty of Veterinary Medicine, University of Sulaimanyah, Sulaimanyah, Kurdistan Region, Iraq

[6] Faculty of Pharmacy, MAHSA University, Jenjarom, Malaysia

Corresponding author
Noorjahan Banu Alitheen,
noorjahan@upm.edu.my

## ABSTRACT

The nanoparticle as a cancer drug delivery vehicle is rapidly under investigation due to its promising applicability as a novel drug delivery system for anticancer agents. This study describes the development, characterization and toxicity studies of a nanostructured lipid carrier (NLC) system for citral. Citral was loaded into the NLC using high pressure homogenization methods. The characterizations of NLC-citral were then determined through various methods. Based on Transmission Electron Microscope (TEM) analysis, NLC-Citral showed a spherical shape with an average diameter size of $54.12 \pm 0.30$ nm and a polydipersity index of $0.224 \pm 0.005$. The zeta potential of NLC-Citral was $-12.73 \pm 0.34$ mV with an entrapment efficiency of $98.9 \pm 0.124\%$, and drug loading of $9.84 \pm 0.041\%$. Safety profile of the formulation was examined via *in vitro* and *in vivo* routes to study its effects toward normal cells. NLC-Citral exhibited no toxic effects towards the proliferation of mice splenocytes. Moreover, no mortality and toxic signs were observed in the treated groups after 28 days of treatment. There were also no significant alterations in serum biochemical analysis for all treatments. Increase in immunomodulatory effects of treated NLC-Citral and Citral groups was verified from the increase in CD4/CD3 and CD8/CD3 T cell population in both NLC-citral and citral treated splenocytes. This study suggests that NLC is a promising drug delivery system for citral as it has the potential in sustaining drug release without inducing any toxicity.

## INTRODUCTION

Exploring natural plant products as an avenue to discover novel chemical entities is one of the fastest growing areas of pharmaceutical research, considering that bioactive agents are

pivotal as potential cures for various diseases. There have been more than one thousand plants identified to possess significant anticancer properties (*Cragg & Newman, 2005*). Citral, with the molecular formula $C_{10}H_{16}O$ is an aldehyde that is widely used in perfumes for its citrus scent. Citral has been claimed to possess anti-inflammatory effects, as it inhibited NO production and suppresses the activation of NF-kappa B in RAW264.7 cells (*Jeong et al., 2008*). Treatment with citral also decreased cell proliferation and altered the mitochondrial membrane potential resulting in apoptotic induction of HeLa and ME-180 cervical cancer cell lines (*Ghosh, 2013*). Together, these versatile characteristics of citral have attracted researchers to further exploit its increased efficiency. However, despite its multi-beneficial effects citral remains an unstable molecule in acidic environments and non-soluble in water (*Weerawatanakorn et al., 2015*). Therefore, the incorporation of citral into colloidal dispersions such as nano-emulsions in particular, could be the best way to stabilize the poorly-soluble compound.

To increase the body fluid saturation solubility of poorly soluble compounds, a new delivery system needs to be developed. The interest of using nanoparticles as a drug delivery system is aimed to improve the bioavailability of drugs, increase the release time in the human body, targeting the drug to its site of action to introduce its physiochemical stability of the drug (*Mudshinge et al., 2011*). Different types of nanocarriers, including lipid nanoparticles have been developed over time for this purpose. Lipid nanoparticles have many desirable features which includes low toxicity, a biodegradable matrix, high capacity to incorporate lipophilic and hydrophilic drugs, and controlled release properties of incorporated drugs (*Kasongo et al., 2011*). The nanostructured lipid carrier (NLC), a second generation of Solid Lipid nanoparticle (SLN)confers all these properties and holds great promise to become an excellent drug carrier system (*Pardeshi et al., 2012*). NLC has greater flexibility for drug loading and exhibits a reduced drug expulsion of the formulation over time that contributes towards long term stability and better modulation of drug release (*Almeida et al., 2014*) as compared to SLN. NLCs are considered as one of the most versatile generation of lipid nanoparticle as oral drug delivery systems to treat cancer. Studies has reported that small particles of NLCs possessing a particle size less than 200 nm have reduced uptake by the Kupffer cells in the liver and prolonged circulation time in blood, allowing better uptake by tumors (*Montenegro et al., 2011*). Additionally, NLC appears to be an attractive approach for the delivery of lipophilic cancer drugs such as Tamoxifen and 5-Fluorouracil (*Andalib et al., 2012*; *How et al., 2013*). With NLC, the drug is expected to solubilize into the hydrophobicphase incorporated into the core of the solid lipid. This could enhance the loading capacity, entrapment efficiency and also control the drug release from the NLC in time. Thus far, there exist no reports which disclose the use of nanostructured lipid carrier as an efficient carrier system to increase the solubility of citral and the physicochemical properties.

One of the major concerns with a nanoparticle system is its inherent toxicity. The safety profiles of citral has been studied on F344/N rat and B6C3F1 mice for two years and been concluded as a non-carcinogenic compound (*Ress et al., 2003*). The potential success of a nanoparticle system relies on many factors, with one of it the minimum toxicity effects exhibited by the carrier itself. Studies have reported that NLC induces no toxicity and no

mortality in an acute toxicity study using BALB/c mice (*Rahman et al., 2014*). Beside the extensive applicability of lipid-based nanoparticle, the toxicity of this system has not been adequately studied thus far. In this present study, combination of hydrogenated palm oil as a solid lipid whilst olive oil a liquid lipid was chosen because this combination was tested to be the least cytotoxic on normal BALB/C mice and 3T3 cells (*How et al., 2013*).

As a potential drug carrier, it is essential to examine the safety and toxicity of citral incorporated into the NLC system to ensure its suitability for the future therapeutic usage of the formulation. For these reasons, the aim of this present work was to determine the physicochemical properties of NLC-citral and to assess the toxicological effects in BALB/c mice treated with NLC-citral.

## MATERIALS AND METHOD

The materials used to synthesize the Nanostructured Lipid Carrier in this project were received from the Institute of Bioscience, UPM without any alteration in the processing method. Hydrogenated palm oil (HPO) was a gift from Institute of Bioscience (Selangor, Malaysia), Lipoid S-100 (Merck Millipore, Germany), and Olive oil (Basso, Italy). Thimerosal, D-Sorbitol, Tween-80, citral 95%, Dulbecco's Modified Eagle's Medium (DMEM), fetal bovine serum (FBS), Thiazolyl Blue Tetrazolium Bromide were all purchased from Sigma-Aldrich (St. Louis, MO, USA). All of the reagents used were of analytical grade except for olive oil.

### Preparation of nanostructured lipid carriers and loading of Citral

The NLCs were prepared using a high-pressure homogenization technique. The synthesis method of NLC loaded with citral was similar to a previous study with slight modifications (*Rahman et al., 2013*). The lipid phase was composed of hydrogenated palm oil (HPO), Lipoid S-100 and Olive oil at a ratio of 7:3:3 that were mixed in a beaker and heated at 70 °C. Five hundred mg of citral was added into the lipid fraction with constant stirring (1,000 rpm) for 5 min. Complete yellowish in color mixture of drug was ensured before proceeding with the next step. For aqueous phase, D-Sorbitol (4.75% [w/v]), Thimerosal (0.005% [w/v]), and 1% Tween-80 were dissolved in double-distilled water at 70 °C and was then added into the lipid phase under constant stirring for 5 min. Further mixing was done by using a high speed stirring method with Ultra-Turrax® (IKA, Staufen, Germany) at 13,000 rpm for 10 min. The pre-emulsion mixture was then pressurized using a high pressure homogenizer (Avestin, Ottawa, ON, Canada) at 1,000 bar for 15 cycles at 70 °C. Finally the hot and clear nanoemulsion was sealed and allowed to cool down at room temperature (27 °C) for 24 h to recrystallize and form the NLCs. The blank NLC was formulated exactly with the same method but without the addition of citral.

### Zeta potential (ZP)

The magnitude charges between the particles in the NLC were determined by using Zetasizer Nano ZS (Malvern, Herrenberg, Germany). NLC was diluted 1:1 ratio with double distilled water appropriately prior to measurements to get optimum of 50–200 kilo counts per second (*Thatipamula et al., 2011*).

### Particle size

The average particle size diameter and polydispersity index (PDI) of the NLC were analyzed by using dynamic light scattering (DLS) integrated in a Zetasizer Nano ZS (Malvern Instrument, Germany). The NLC solution was diluted at 1:1 ratio with double distilled water preceding the measurement. Five independent measurements were obtained at 25 °C with particle size and PDI (size distribution) were calculated.

### Transmission electron microscopy (TEM)

The NLC-Citral sample was dropped slowly on the surface of a copper grid coated with carbon and then let to air dry for about 5 min. Negative staining was performed by using 1% phosphotungstic acid then allowed for air drying for another 3 min. The sample was then visualized with transmission electron microscopy (Hitachi H-7100, Japan).

### Entrapment efficiency (EE) and drug loading (DL) capacity

The amount of free drug in the sample was measured in order to determine the entrapment efficiency and the drug loading capacity of the sample by using ultrafiltration method. This was performed using Centisart filter tubes (Sartorious, Goettingen, Germany) with molecular weight cut off 300 kDa. Five mL of NLC-citral was placed in the upper chamber of the tubes and then was centrifuged using Eppendorf Centrifuge (Hamburg, Germany) at 15,000 rpm for 15 min. The foundation behind this technique is that citral filtered out to the bottom chamber is unbound to the NLC while the citral remained in the top chamber were still bound as NLC-citral. Subsequently, citral in the bottom chamber was quantified using UV-Vis spectrophotometer at 280 nm (Beckman-Coulter, Fullerton, CA, USA). The EE and DL percentage of the sample were calculated based on the method previously reported (*Rahman et al., 2013*).

$$EE(\%) = \frac{(\text{Total amount of citral}) - (\text{Free amount of citral})}{(\text{Total amount of citral})} \times 100$$

$$DL(\%) = \frac{(\text{Total amount of citral encapsulated into NLC})}{(\text{Total amount of lipid used in NLC-citral formulation})} \times 100.$$

### *In vitro* drug release study

The *in vitro* drug release study of NLC-citral was accomplished using a Franz diffusion cell system (PermeGear, Hellertown, PA, USA) with a receptor volume of 20.1 cm$^3$ and effective diffusion area of 4.9 cm$^2$. This study is essential to determine the suitability of the drug in the system and to check the quality of the formulation. There were donor and receptor compartments in the system with donor cell having the surface area of 20.1 cm$^2$. The release of the NLC-citral was studied for 48 h. Firstly, the synthetic cellulose acetate membrane (Advantec, Tokyo, Japan) with pore size of 200 nm was soaked in the receiving media for 2 h. Next, the membrane was inserted onto the system and 500 µL of NLC-citral suspension were dispensed in the donor compartment (*Salerno, Carlucci & Bregni, 2010*). The receptor compartment was filled with the receiving medium (PBS + 2% SDS) at pH 7.4. Throughout the study, the temperature of the system was maintained at 37 °C under constant stirring. At certain time intervals, 500 µL of sample was withdrawn from the medium in receptor

compartment by using a 3 mm$^3$ syringe needle and then refilled back by using the fresh receiving medium. The samples were then analyzed by UV-Vis spectrophotometer at 280 nm (Beckman-Coulter, Fullerton, CA, USA) to determine the amount of citral released from the NLC. The concentration of drug release was calculated based on the standard curve for pure citral ranging from 100 µg/mL to 10 µg/mL. The experiment was conducted in triplicate and the drug release data were evaluated by zero-order, first-order, and Higuchi equations to determine the release kinetics (*Rahman et al., 2013*).

### *In vitro* splenocyte viability (MTT)

In brief, upon mice sacrifice, the spleen collected was washed with 5 mL of PBS. It was then lysed with 2 mL lysis buffer (155 mM$NH_4Cl$, 0.1 M $Na_2EDTA$, 12 mM $KH_2PO_4$ at pH 7.4) for 10 min to remove the red blood cells. Next, the splenocytes were seeded at a density of $2 \times 10^6$ cells/mL in a 96-well plate. Cells were treated with seven different concentrations: NLC-Blank, NLC-citral and citral starting from 0.8 µg/mL to 30 µg/mL for 24 h of incubation in a 37 °C incubator with 5% $CO_2$ (Memmert, Schwabach, Germany). Then, 20 µL of MTT3-[4,5-dimethylthiazol-2-yl]-2,5 diphenyltetrazolium bromide (5 mg/mL) (Merck, Kenilworth, NJ, USA) solution was added into the cells and incubated for another 4 h in the incubator. The plate was spun down at 1,500×g for 5 min. Next, the supernatant was removed and 100 µL of dimethyl sulfoxide (DMSO) (Fisher Scientific, Hampton, NH, USA) was added in to the plate and incubated for another 15 min at room temperature. The plate was read with microplate reader at 570 nm. *In vitro* splenocytes viability assay was done according to the previous study (*Abu et al., 2015*). The percentage of cells viability was calculated based on the formula below:

$$\text{Viability}(\%) = \frac{\text{OD of sample}}{\text{OD of control}} \times 100.$$

### Animal

Male BALB/c male mice aged seven weeks (20–22 g) were purchased from the Animal House of Faculty of Veterinary, Universiti Putra Malaysia (UPM, Malaysia). This study was approved by the Institutional Animal Care and Use Committee, Universiti Putra Malaysia (R098/2014). The mice were randomly selected and grouped into 3 groups with five mice per group; NLC-Blank, NLC-citral and citral. Fifteen mice were acclimatized in the laboratory environment for 7 days. The mice were accommodated in polypropylene plastic cage with pellet and water provided *ad libitum* during the period of study. The treatment of 50 mg/kg/day of NLC-citral, citral and NLC-Blank were administered orally for 28 days following the method described earlier (*Mohamad et al., 2015*). The animals were observed for any toxic signs such as loss of weight, shedding of fur or behavioral abnormalities after 28 days of study.

### Serum biochemistry

Blood samples were collected in a plain tube from the sacrificed mice by cardiac puncture. The serum was obtained by centrifugation at 3,000 rpm for 15 min. The concentration of aspartate aminotransferase (AST), alanine aminotransferase (ALT), alkaline phosphatase
**Table 1   The summary of characterizations for NLC-Citral.**

| Element | Value |
|---|---|
| Zeta potential | $-12.59 \pm 0.52$ mV |
| Zeta sizer | $54.12 \pm 1.10$ nm |
| Polydispersity index | $0.224 \pm 0.005$ |
| Entrapment efficiency | $98.9 \pm 0.124\%$ |
| Drug loading capacity | $9.84 \pm 0.041\%$ |

(ALP) and creatinine level in mice serum were analyzed accordingly using Hitachi automatic analyzer (Hitachi-902, LTD, JAPAN).

### *In vivo* immunophenotyping of splenocytes

Mice spleen obtained from the sacrificed mice were washed with Phosphate Buffered Saline (PBS) and meshed through a 40 μm sterile filter with plunger. Next, the suspension was washed with PBS and centrifuged at 2,000×g for 15 min to get the cells pellet. Then, the red blood cell was lysed with 2 mL of lysis buffer (155 mM $NH_4Cl$, 0.1 M $Na_2EDTA$, 12 mM $KH_2PO_4$ at pH 7.4) and incubated for 10 min in 4 °C. After that, the suspension was again spun at 2,000×g for 5 min. Cell was resuspended with PBS for counting purposes. About $1 \times 10^6$ cells/mL was stained with antibodies (CD3, CD4, CD8) (Abcam, San Francisco, CA, USA) for 2 h with constant shaking. Then, the cells were fixed in 600 μL of 4% paraformaldehyde (Sigma, USA) before being subjected to flow cytometry analysis (FACS Calibur, BD, San Jose, CA, USA).

### *In vivo* nitric Oxide detection of the spleen

Nitric Oxide detection from the spleen was established with the Griess Reagent Kit (Sigma, USA). The experiment was conducted according to the protocol provided with the kit. The supernatant from the splenocytes was mixed with Griess Reagent and incubated for 30 min before being measured at 548 nm using microplate reader (Beckman Coulter, USA).

### Statistical analysis

The obtained data statistically analyzed. All of the results were expressed as the means and standard deviations. The statistical analysis was performed using the GraphPad Prism (Version 6.0). Statistical significance ($p < 0.05$) was assayed by one-way ANOVA analysis.

## RESULTS AND DISCUSSION

### Physicochemical characterisation of NLC-citral formulation
#### *Particle size and polydispersity index*

NLC-Citral was successfully synthesized using the high pressure homogenization method. The summary of physicochemical characteristics of NLC-citral formulation is as shown in Table 1. Particle size is the most fundamental parameter in a nanoparticle study. The particle size and polydispersity index is highly important in determining the physical stability of colloidal dispersion system of the nanoparticle (*Lee, Lim & Kim, 2007*). The mean particle size and polydispersity index of NLC-citral were obtained as $54.12 \pm 0.30$ nm and $0.224 \pm 0.005$, respectively.

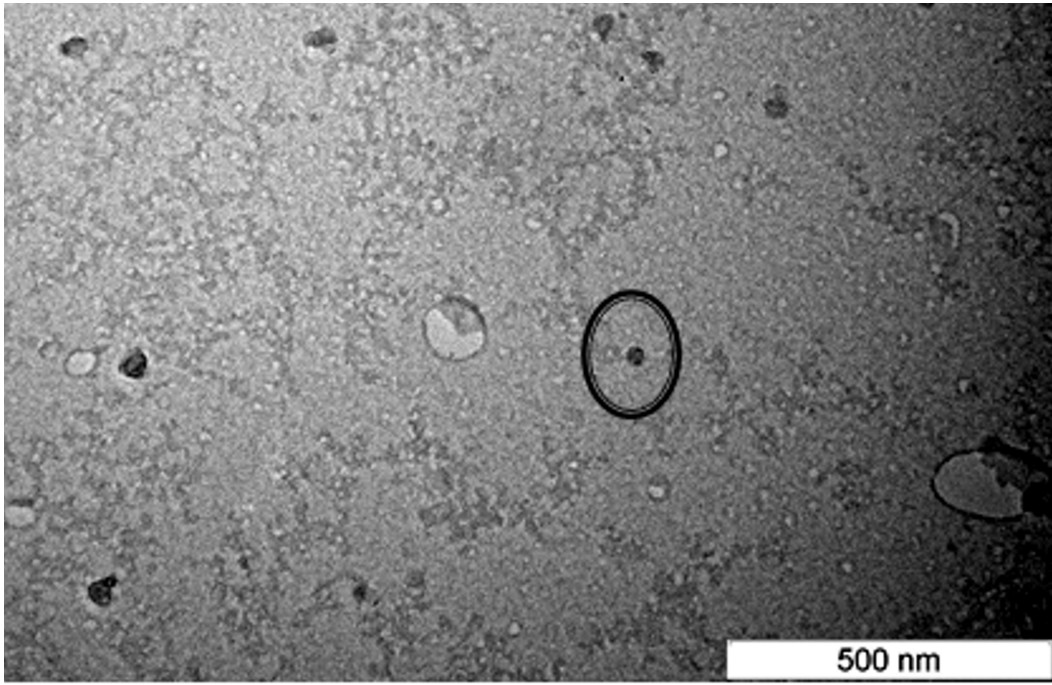

500 nm

**Figure 1** **Transmission electron microscope (TEM) image of the NLC-Citral prepared by negative staining at 80,000X magnification.** The circle represents the nanoparticle which can be seen in regular circle shape.

Particle size of the nanoparticle was present in the average diameter of the nanoparticle. It is crucial to confirm that the size of the nanoparticle formulated is within nano-sized range (*Mukherjee et al., 2008*). As depicted in Fig. 1, NLC-citral was observed to confer regular shapes and are relatively uniform suggesting good particle dispersity. There was a minor difference in size distribution data obtained from TEM and zeta sizer. This small disparity in measurement was probably due to differences in the techniques and method of data analysis between both instruments. The measurement of nanoparticles is mostly influenced by the conditions, method of analyzing data and also depends on the operation of instrument in the determination (*How et al., 2013*). From this result, it was concluded that this formulation has fallen within a nano-sized range. The size of the particle plays a key role towards their adhesion to and interaction with biological cells (*Foster, Yazdanian & Audus, 2001*). Being a small nanoparticle, NLC-Citral holds a potential in lowering the risk of non-specific liver uptake (*Seki et al., 2004*) as well as increasing the actual distribution and bioavailability of the drug (*Fang et al., 2006*). The incorporation of a water insoluble anticancer compound (Amoitone B) with NLC was reported to enhance its bioavailability (*Luan et al., 2013*). Moreover, small nanoparticle with a diameter size of 50–60 nm resulted in high tumor uptake as well as sustained release of the drug in cancer biology studies (*Mitra et al., 2001*; *Sharma et al., 1996*).

### Zeta potential

Determination of zeta potential for NLC-citral was done to evaluate the overall stability of the formulation. The average zeta potential for NLC-citral was $-12.73 \pm 0.34$ mV (Table 1). Stability test for a nano-suspension is very crucial in any nanoparticle studies. Zeta potential is one of the parameters that govern the stability of a nanoparticle formulation. As a rule of thumb, zeta potential ranging from $-5$ mV to $+5$ mV indicates low stability and higher possibility for aggregation (*Honary & Zahir, 2013*). Generally, a zeta potential value must be higher than $+30$ mV or lower than $-30$ mV (*Rahman et al., 2013*). Surfactants were used as a steric stabilizer during the nanoparticle formulation to control particle size and the stability of dispersion. Surfactants have a wide range of uses in pharmaceutical preparations. In this formulation, Tween 80 was chosen to provide a good steric stabilization. Previous finding has proved that Tween 80 can compensate the stability of NLC and SLN dispersions by providing a steric stability in Domeperidone loaded with NLC and SLN (*Thatipamula et al., 2011*). Conversely, studies have concluded that, nanosuspension with slightly low zeta potential is stable enough with the additional usage of higher molecular weight of stabilizer which will act by steric stabilization in the particle (*Honary & Zahir, 2013*; *Lee, Lim & Kim, 2007*). Hence, the stabilizer used in the formulation of nanoparticle effected the average size and charges of the NLC (*Abdelwahab et al., 2013*).

### Entrapment efficiency and drug loading capacity

The entrapment efficiency study is required to determine the high performance from the formulation. In this formulation of NLC, Citral conveyed $98.9 \pm 0.124\%$ of entrapment efficiency in the system with $9.84 \pm 0.041\%$ of drug loading capacity in the lipids concentration. Incorporation of citral has led to a high percentage of entrapment efficiency (EE), probably because of its lipophilicity and low water solubility characters. High EE is required for good encapsulation parameters in any drug nano-carrier system including NLC. Furthermore, the amount of drug entrapped in the nano-carrier also determines the performance of drug delivery system since it influences the rate of the drug release from the system (*Tiwari et al., 2012*). The addition of drugs in the NLC system minimizes the interfacial retention in between lipid matrix and liquid phase whereby reduces the free energy within phase boundary of lipid and the drug incorporated (*How et al., 2013*). Ideally a high loading capacity is aimed because it would improve the entrapment efficiency of the drug eventually (*Patel et al., 2012*). Drug loading is defined as the process of incorporation of the drug into the carrier system; meanwhile entrapment efficiency portrays the effectiveness of the incorporated drug into the carrier. The solubility of the drug in lipid used assures the maximum drug loading in the NLC system (*Rahman et al., 2013*). Hence, high loading capacity of the citral in NLC system examined was due to the good solubility of citral in the lipid phase of the NLC. The formation of unordered lipid crystal in the NLC has increased the loading capacity. Imperfect crystal formation in the lipid nanoparticle can be achieved through the mixing of solid lipid with liquid lipid which would lead to a higher drug loading of NLC (*Müller, Radtke & Wissing, 2002*).

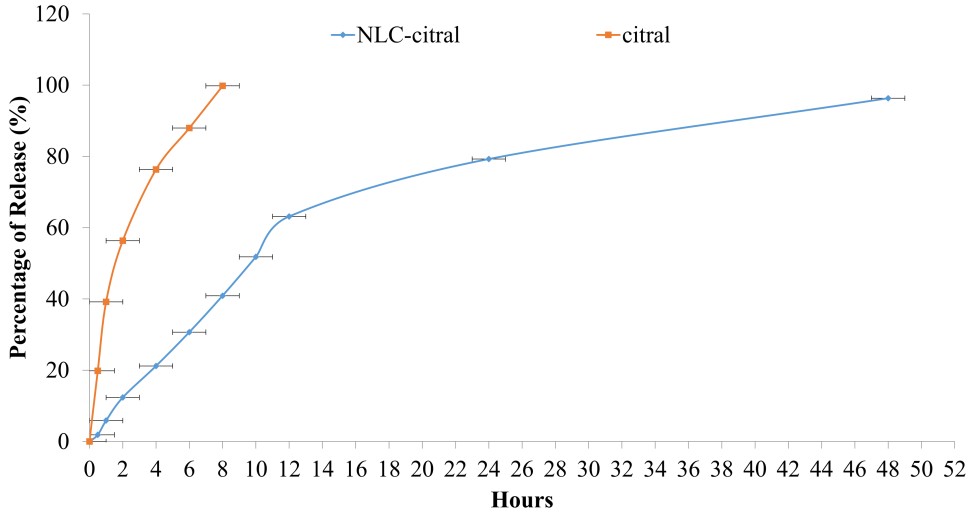

**Figure 2** **The *in-vitro* drug release study of NLC-citral and citral (control).** Each point represents mean and standard deviation ($n = 3$).

## Drug release study

Furthermore, the drug release profile of NLC-citral was also determined in this study. The cumulative percentage of citral from NLC-citral over 48 h is as shown in Fig. 2. *In vitro* drug release study was performed using the Franz diffusion cell system in order to determine the release profile of the citral from NLC-citral for 48 h. NLC system has shown a constant release rate of citral release with a total of 96.3 ± 2.1% after 48 h of study.

It has become an essential strategy that drug delivery system should possess a better drug release capacity to improve its effectiveness and reduce side effects of the drug caused by a rapid dosage consumption by the patients in a therapeutic routine (*How et al., 2013*). It was observed that a slow release profile of citral from NLC was apparent, compared to the administration of citral alone (Fig. 2). This study also showed that the drug release kinetics of citral from NLC followed a zero-order kinetic model, with an $r^2 = 0.976$. Relevant to the objective of the study, NLC proved to be a suitable carrier of citral as it provides a slow release of it in time dependent manner. A biphasic drug release pattern was observed in which a drug burst release was observed at the initial stage and followed by sustained release at a constant rate later on. This phenomenon might be explained due to the lipid imperfect crystalline structured of the NLC to allow high drug loaded in the system (*Müller, Radtke & Wissing, 2002*). Subsequently, this also affects the release of the drug as it becomes loose and causing high rate of the drug released (*Uprit et al., 2013*). In addition, as the size of the nanoparticle is getting smaller, the specific surface area of nanoparticle is increased and thus the drug release became faster (*Moghimi, Hunter & Murray, 2005*). Therefore, the robust release rate in the initial stage of this in-vitro drug release study was resulted from these two consequences. Consequently, citral in NLC-citral showed slower release profile. Previous study has confirmed that the slow release of the drug was dependent on

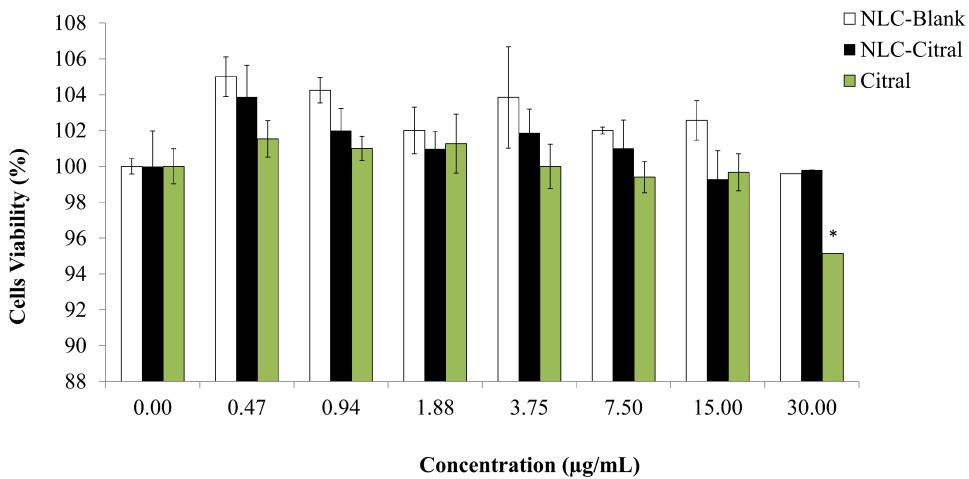

**Figure 3** **The percentage of splenocytes viability after being treated with various concentrations of NLC-citral and pure citral ranging from 0.47 µg/mL to 30 µg/mL for 24 h.** Each point is represent the mean ± standard deviation ($n = 3$). Significance was set at $p < 0.05$ comparing between groups with (**) to NLC-Blank and (*) to citral.

the entrapment efficiency and homogenous entrapment of the drug throughout the system (*Moghimi, Hunter & Murray, 2005*).

## *In vitro* splenocytes viability (MTT)

To examine the effect of splenocytes proliferation after being treated with NLC-Citral formulation, the MTT assay was conducted. The *in vitro* proliferation response of splenocytes treated with NLC-citral and citral was investigated and showed in Fig. 3. There was no significant decline in the viability of splenocytes detected from treatment with NLC-citral (99.8 ± 1.7%) at the highest concentration given to the mice (30 µg/ml) after 24 h post-treatment as compared to NLC-Blank (99.5 ± 0.98%) which suggests that the formulation did not induce toxicity towards the proliferation of mice splenocytes after being treated (*Chen et al., 2009*). According to Fig. 3, the percentage of splenocytes viability did not change in the treatment of NLC-citral at any concentration given. However, in the citral treated group there was a decrease in splenocytes viability from 99.7% to 95.2% at the concentration of 15 µg/mL and 30 µg/mL respectively.

It was demonstrated that the NLC-Citral with several concentrations did not alter the effects of splenocytes proliferation treated. As a whole, NLC-Citral has the beneficial effect on immune-stimulation as the splenocytes proliferation was more noticeably better for NLC-citral treated than citral treated group. Immune cells such as T and B cells play an important role in the host defense system. Splenocytes proliferation indicates the ability to increase numbers of immune markers which could stimulate the host immunity (*Rescigno, Avogadri & Curigliano, 2007*).

## *In vivo* toxicity study

All mice treated with NLC-citral and citral survived till the end of the 28 days of treatment. According to Table 2, there were no mortality, body weight changes, and conspicuous

**Table 2   The observation of mortality, body weight changes, toxicity signs and serum biochemical analysis of NLC-Blank, NLC-Citral and Citral treated groups.** Values represent means and standard deviation.

| Group | Mortality at 28th days | Body weight changes | Toxic signs | ALT (U/L) | ALP (U/L) | AST (U/L) | CREATININ ($\mu$mol/L) |
|---|---|---|---|---|---|---|---|
| NLC-Blank (Control) | NONE | NONE | NONE | 75.2 ± 0.37 | 230 ± 0.65 | 348.2 ± 1.56 | 49.0 ± 1.09 |
| NLC-CITRAL | NONE | NONE | NONE | 76.1 ± 1.03 | 225 ± 0.98 | 351.9 ± 0.98 | 47.0 ± 0.97 |
| CITRAL | NONE | NONE | NONE | 71.5 ± 0.76 | 232 ± 1.34 | 359.0 ± 0.56 | 47.6 ± 1.0 |

toxic signs developed in treated mice throughout the study time. For biochemical analysis, there was a non-significant increase in the serum ALT level of NLC-citral treated group as compared to NLC-Blank. Meanwhile, the citral treated group showed a decrease in ALT value. Besides, creatinine level of the NLC-citral and citral treated groups has decreased than NLC-Blank.

This observation indicates that there was no overt toxicity occurred in NLC-citral and citral treated mice. Even though slight changes were observed in the level of markers from the biochemistry results, there was no significant modification detected. Hepatic damage is a serious issue that should be taken into consideration in the new formulation of compound or drugs. Additional AST enzyme will be released into the bloodstream when liver is damaged that indicates a hepatic injury in the host (*Sharma & Shukla, 2011*). ALT level is a main indicator in the detection of liver damage by drug or hepatotoxins in the bodies (*Shirodkar et al., 2015*). Eventually, the increase is AST level caused an elevation in the level of ALT (*Giannini, Testa & Savarino, 2005*). Moreover, the abnormal levels of ALP, AST and ALT in the serum most often indicate a problem with the liver, gall bladder and heart (*Shimizu, 2008*). In addition, creatinine level was assessed to view the renal function, an elevation of the creatinine level in the blood signifies impaired kidney function or kidney disease (*Shlipak et al., 2013*). There were no significant alterations in the levels of kidney and renal biochemical detector molecules in the NLC-Citral and Citral treated groups of mice. Pharmacology study has claimed that Citral is not toxic for human consumption as indicated by no changes and abnormalities in serum glucose, urea, proteins and creatinine level in the urine analysis (*Carlini et al., 1986*).

### *In vivo* nitric oxide detection from the spleen

NO level from the NLC-citral and citral treated groups was evaluated to study the anti-inflammatory effects of the formulations. As illustrated in Fig. 4, both NLC-citral and citral have significantly reduced the level of NO. The level of nitric oxide in the NLC-citral group was declined from 101 ± 3.02 $\mu$M/mg (NLC-Blank) to 40.91 ± 3.02 $\mu$M/mg of protein while in citral the value was dropped to 46.16 ± 4.025 $\mu$M/mg of protein.

Nitric Oxide (NO) is a signaling molecule that initiates inflammation under normal conditions. High concentrations of NO in the immune system was relatively regulated by cytokine-activated macrophage (*Sharma, Al-Omran & Parvathy, 2007*). The level of NO in the NLC-citral treated spleen has significantly reduced as compared to the NLC-Blank. Chronic inflammation facilitates cancer-related potentials such as inhibition of cell death

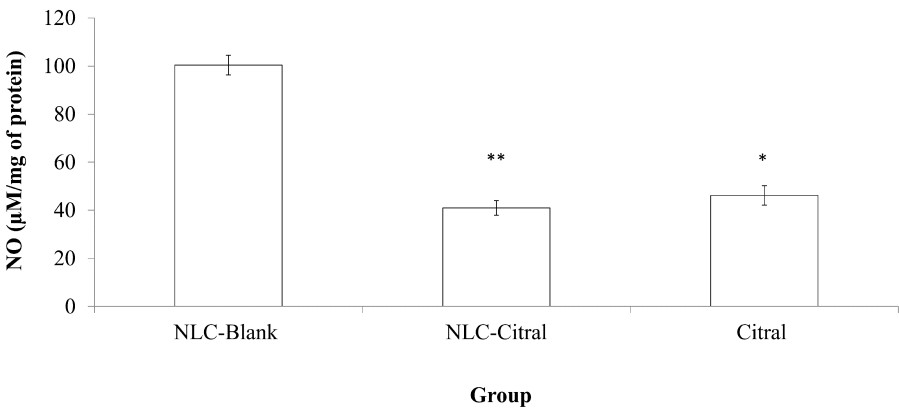

**Figure 4** Bar chart analysis of the level of Nitric Oxide (NO) detected in the mice splenocytes harvested from the NLC-Blank, NLC-citral and citral treated groups after 28 days of treatment (50 mg/kg). Each value represents the means ± standard deviation. Significance was set at $p < 0.05$ comparing between groups with (*) to NLC-Blank and (**) to citral.

programs, induce angiogenesis and assists metastasis (*Multhoff, Molls & Radons, 2011*). Hence, the reduction NO level in NLC-citral depicts the anti-inflammatory effect possessed in the formulation.

## Splenocytes immunophenotyping

To determine the immune response affected from the NLC-citral and citral treatment as compared to the NLC-Blank, *in vivo* immunophenotyping assay was done. Three different conjugated staining of CD3 (FITC), CD4 (PE) and CD8 (APC) T-cells immune markers were selected to stain the harvested splenocytes from all groups. As shown in Fig. 5, the percentage of CD4/CD3 T-cell population has significantly increased in the NLC-citral and citral treated groups as compared to the NLC-Blank. The level of CD4/CD3 T-cells population increased from 13.97 ± 0.19% in NLC-Blank to 20.12 ± 0.40% in the NLC-citral treated splenocytes and 18.44 ± 0.50% in the citral treated splenocytes (50 mg/kg). Furthermore, the level of CD8/CD3 T-cell population was also increased, similar to the pattern of elevation observed in the CD3/CD4 population. It can be observed from Fig. 5 that the percentage of CD8/CD3 T-cells population in the NLC-citral and citral treated splenocytes has significantly increased to 9.56 ± 0.32% and 8.82 ± 0.16% respectively.

NLC-citral has increased the percentage of T-cells population higher than citral for all three T-cells immune markers as compared to the NLC-Blank within the splenocytes population. Previous study reported that citral exhibited potential as an immunomodulatory agent in murine macrophage (*Bachiega & Sforcin, 2011*). Therefore, the increase of CD4/CD3 and CD8/CD3 in both NLC-citral and citral treated splenocytes conveyed that these treatments have induced the immunomodulatory effect of the mice. Studies on the immune responses have continued to gain significant attraction due to the importance of immune system in combating against common critical diseases such as cancer (*Dunkelberger & Song, 2010*). T-cell immune response impairments and dysfunction almost appear in many chronic diseases such as cancer and Human immunodeficiency

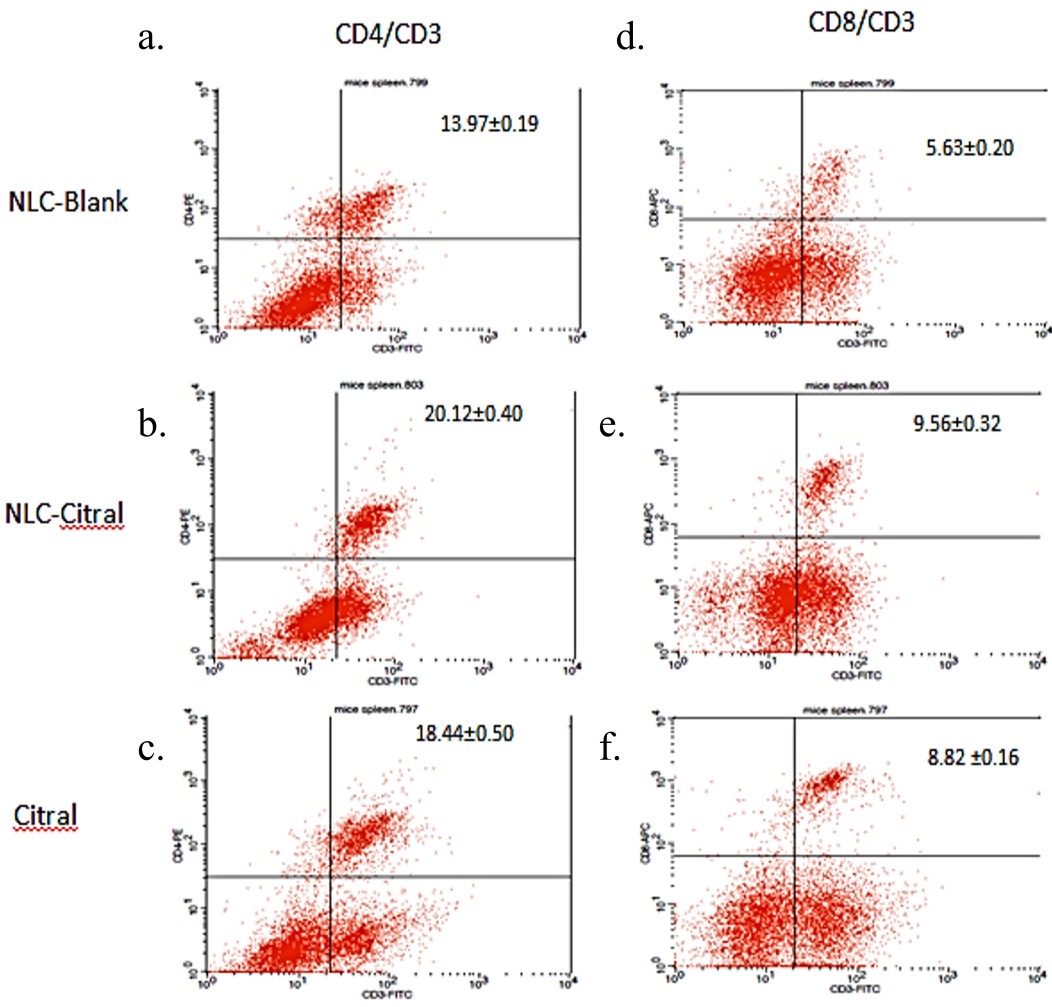

**Figure 5** The histogram analysis of (A–C) CD3$^+$CD4$^+$ and (D–F) CD3$^+$CD8$^+$ T lymphocytes in the splenocytes harvested after 28 days of treatment with 50 mg/kg of NLC-Blank, NLC-citral and citral. Each value represents mean ± std with $n = 5$ mice per group. (A) indicates CD3$^+$CD4$^+$ T lymphocytes populations upon treatment with NLC-Blank; (B) indicates CD3$^+$CD4$^+$ T lymphocytes populations upon treatment with NLC-citral; (C) indicates CD3$^+$CD4$^+$ T lymphocytes populations upon treatment with citral; (D) indicates CD3$^+$CD8$^+$ T lymphocytes populations upon treatment with NLC-Blank; (E) indicates CD3$^+$CD8$^+$ T lymphocytes populations upon treatment with NLC-citral; (F) indicates CD3$^+$CD8$^+$ T lymphocytes populations upon treatment with citral.

virus (HIV) infection disease (*Kinter et al., 2004*). CD4/CD3 is the T-helper lymphocyte cells while CD8/CD3 is the cytotoxic T cells (*Anderson, Blue & Schlossman, 1988*). The depletion of T-helper cells creates an immune hyper activation conditioned which is usually associated with the immune response deficiency (*Février, Dorgham & Rebollo, 2011*). In addition, a cytotoxic T lymphocyte cell CD8/CD3 leads to the death of target cells by apoptosis in host defense against various pathogens (*Janeway et al., 2001*). Much attention has focused on the role of CD8T-cells in immunotherapy of cancer as it is capable to destroy tumor cells in *in vitro* and *in vivo* studies (*Finn, 2003*).

## CONCLUSION

In the present study, the formulation of nanostructured lipid carrier encapsulated with citral (NLC-citral) was characterized. The formulation was confirmed to be stable and fell within the nano size range. This was proved based on the results from zeta potential, zeta sizer, polydispersity index and TEM analysis of the nanoparticle. Furthermore, NLC-citral formulation imposes a slow release profile of citral and has enhanced the solubility of the pure citral in the water. Additionally, there was no aberrant toxic sign detected in the blood biochemistry analysis, flow-cytometry immunophenotyping assay, lipid peroxidation (NO) and splenocytes viability between the NLC-Blank (control) and NLC-citral and citral treated mice in the toxicity study conducted. The physicochemical properties have validated that NLC-citral is suitable as a potential delivery system of citral and non-toxic towards healthy cells. Hence, the NLC-citral formulation can be further investigated to confirm its potential as a new delivery system for citral in the treatment of cancer as citral is known to have anti-cancer properties.

**List of abbreviations**

| | |
|---|---|
| **ALP** | Alkaline phosphatase |
| **ALT** | Alanine aminotransferase |
| **APC** | Antigen-presenting cells |
| **AST** | Aspartate Aminotransferase |
| **ATCC** | Animal Tissue Culture Collection |
| **CDK** | Cyclin Dependent Kinase |
| **DL** | Drug Loading |
| **DMEM** | Dulbecco's Modified Eagle Medium |
| **DMSO** | Dimethyl sulfoxide |
| **EDTA** | Ethylenediaminetetraacetic acid |
| **EE** | Entrapment Efficiency |
| **FACS** | Fluorescence-activated cell sorter |
| **HPO** | Hydrogenated Palm Oil |
| **MTT** | 3-(4,5-dimethylthiazol-2-yl)-2,5-diphenyltetrazolium bromide |
| **NaCl** | Sodium chloride |
| **NO** | Nitric oxide |
| **PBS** | Phosphate buffer saline |
| **PI** | Propidium Iodide |
| **TEM** | Transmission Electron Microscope |
| **WHO** | World Health Organization |

### Funding

This study was supported by Research University Grant Scheme (Project No: 05-02-12-1718RU) provided by Universiti Putra Malaysia, Malaysia. The funders had no role in study design, data collection and analysis, decision to publish, or preparation of the manuscript.

## Grant Disclosures

The following grant information was disclosed by the authors:
Research University Grant Scheme: 05-02-12-1718RU.

## Competing Interests

The authors declare there are no competing interests.

## Author Contributions

- Noraini Nordin conceived and designed the experiments, performed the experiments, analyzed the data, wrote the paper, prepared figures and/or tables.
- Swee Keong Yeap conceived and designed the experiments, analyzed the data, reviewed drafts of the paper.
- Nur Rizi Zamberi performed the experiments.
- Nadiah Abu reviewed drafts of the paper.
- Nurul Elyani Mohamad, Heshu Sulaiman Rahman and Chee Wun How analyzed the data.
- Mas Jaffri Masarudin analyzed the data, reviewed drafts of the paper.
- Rasedee Abdullah contributed reagents/materials/analysis tools, reviewed drafts of the paper.
- Noorjahan Banu Alitheen conceived and designed the experiments, analyzed the data, contributed reagents/materials/analysis tools, reviewed drafts of the paper.

## Animal Ethics

The following information was supplied relating to ethical approvals (i.e., approving body and any reference numbers):

This study was approved by the Institutional Animal Care and Use Committee, Universiti Putra Malaysia (R098/2014).

## Data Availability

The raw data has been provided as a Supplemental File.

## Supplemental Information

Supplemental information for this article can be found online at http://dx.doi.org/10.7717/peerj.3916#supplemental-information.

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
