# Peer review of "Characterization and toxicity of citral incorporated with nanostructured lipid carrier"

_PeerJ, doi:10.7717/peerj.3916_

## Round 0.1 · original submission · Major Revisions

English language, control experiment, combining results and discussion. Rewriting of some of the sections as suggested by both reviewers are needed.

Reviewer 1 ·

Basic reporting

The article is well written, clear, structured and it is within the field of the journal. The article describes the production of lipid nanoparticles of citral with a good particle size, PI, ZP, EE and biocompatibility. The article is well illustrated. This article can be accepted for publication after a major revision.

Experimental design

The design of the production and caracterization of the NLC prepared is well done.
The methods are well described. The article is well illustrated.

Validity of the findings

The article is well written and it is within the field of the journal. The results and conclusions are acceptable.

Additional comments

The article is well written and it is within the field of the journal. The article describes the production of lipid nanoparticles of citral with a particle size, PI, ZP and good EE. The article is well illustrated. This article can be accepted for publication after a major revision.

Minor Changes:
• Add a list of abbreviations in the end of the article, after the conclusions subchapter (in the abstract have the abbreviation NO?! I know what is but some authors no…. a list of abbreviations solve this problem);

• In the introduction, the authors should highlight the advantages of lipid nanoparticles compared to other colloidal carriers. Explain why you chose the NLC and not SLN! Describe the advantages of this kind of second generation of lipid nanoparticles. Authors can see these in the article: (cite the article)
Almeida H, Amaral MH, Lobão P, Silva AC, Sousa Lobo JM. Applications of lipid and polymeric nanoparticles in ophthalmic pharmaceutical formulations: present and futures considerations. J. Pharm Pharm Sci. 2014; 17(3)278-293.

• Why you chose these solid and liquid lipids to prepare the NLC? Are you sure that are the best choice? Justify!

• In the preparation of NLC detail the rpm of the stirring and time.

• In the determination of EE% describe the wave-length (nm) used in the UV-Vis spectrophotometer. The same in the in vitro dug release study.

• In the in vitro splenocyte viability (MTT) why you use these rage of concentrations of the formulation? Support with bibliography.

• ZP values prove that the optimized formulation is not stable over time. Why you did n´t optimized the formulation or add something to improve this parameter? How you justified this? Improve this because your explanation is not enough…



Major Changes

• In order to improve the quality of the article, the subchapter results and discussion must be united in only one subchapter. It is better for the readers to understand your research.

Reviewer 2 ·

Basic reporting

Manuscript title: Characterization and Toxicity of Citral Incorporated With Nanostructured Lipid Carrier
Authors: Nordin et al
The manuscript describes the physicochemical characterization of the Citral loaded NLC. Works appear to be systematically carried out and trustworthy. The manuscript is worthy to be published. However, some corrections are to be made before a final decision could be made.
The level of English in its present form is not acceptable. Besides there are some typos and merged words. Authors should have made a general correlation between the different experimental data set. Some specific comments are appended below:
1. In the introduction section the authors should make some statement as how the present manuscript is going to add further information in the field of NLC research. What is the novelty of the present work?
2. Abstract: Polydispersity index does not have any unit.
3. Authors should justify such a higher extent of entrapment efficiency (99%). Is it trustworthy?
4. The interpretation on the TEM images (Figure 1) is completely wrong. NLCs are elsewhere, which the authors have failed to mark.
5. Authors have not shown the control experiment. This is absolutely essential to prove that the citral release from the NLC is sustained. Also authors should propose as which model is best fitted to the release kinetics.
6. I failed to follow the meaning of 105% cell viability (Figure 3), especially for the blank NLC. What does it signify?
7. Line 399: statement is wrong. Authors should rather justify as how the systems with lower magnitude of zeta potential is stable. Author should specifically mention that the used Tween provided steric stabilization to the NLC formulations.
8. Figure 3: Authors should comment on the optimum concentration of the formulation.
9. The conclusion section should be rewritten. Some future perspectives are worth mentioning.

Experimental design

Looks fine

Validity of the findings

Alright

---

## Round 0.2 · accepted · Accept

Your revised manuscript is accepted for publication in PeerJ.

Reviewer 1 ·

Basic reporting

I accept the changes done by the authors! The manuscript must be published! Thanks

Experimental design

I accept the changes done by the authors! The manuscript must be published! Thanks

Validity of the findings

I accept the changes done by the authors! The manuscript must be published! Thanks

Additional comments

I accept the changes done by the authors! The manuscript must be published! Thanks